# RealCause: Realistic Causal Inference Benchmarking

**Brady Neal**
Mila, Univerité de Montréal

**Chin-Wei Huang**
Mila, Univerité de Montréal

**Sunand Raghupathi**
Mila, Univerité de Montréal

## Abstract

There are many different causal effect estimators in causal inference. However, it is unclear how to choose between these estimators because there is no ground-truth for causal effects. A commonly used option is to simulate synthetic data, where the ground-truth is known. However, the best causal estimators on synthetic data are unlikely to be the best causal estimators on real data. An ideal benchmark for causal estimators would both (a) yield ground-truth values of the causal effects and (b) be representative of real data. Using flexible generative models, we provide a benchmark that both yields ground-truth and is realistic. Using this benchmark, we evaluate over 1500 different causal estimators and provide evidence that it is rational to choose hyperparameters for causal estimators using predictive metrics.

## 1 Introduction

In causal inference, we want to measure causal effects of treatments on outcomes. Given some outcome $Y$ and a binary treatment $T$, we are interested in the *potential outcomes* $Y_i(1)$ and $Y_i(0)$. Respectively, these denote the outcome that unit $i$ would have if they were to take the treatment ($T = 1$) and the outcome they would have if they were to not take the treatment ($T = 0$). We are often interested in causal estimands such as $\mathbb{E}[Y(1)] - \mathbb{E}[Y(0)]$, the *average treatment effect* (ATE). This is equivalent to the following expression using Pearl's do-notation (Pearl, 1994, 2009, 2019): $\mathbb{E}[Y \mid \mathrm{do}(T = 1)] - \mathbb{E}[Y \mid \mathrm{do}(T = 0)]$, where $\mathrm{do}(T = t)$ is a more mnemonic way of writing that we set the value of the treatment to $t$.

There are many different estimators for estimating causal estimands (see, e.g., Neal, 2020; Hernán & Robins, 2020; Morgan & Winship, 2014; Imbens & Rubin, 2015, and Appendix E). However, it is unclear how to choose between these estimators because the true values of the causal estimands are generally unknown. This is because we cannot observe both potential outcomes (Rubin, 1974), so we have no ground-truth. This is often referred to as the *fundamental problem of causal inference* (Holland, 1986). Supervised machine learning does not have this "no ground-truth" problem because it is only interested in estimating $\mathbb{E}[Y \mid T]$, which only requires samples from $P(Y \mid T)$, rather than samples from $P(Y \mid \mathrm{do}(T = 1))$ and $P(Y \mid \mathrm{do}(T = 0))$. Yet, we must choose between causal estimators. How can we do that when faced with the fundamental problem of causal inference?

To evaluate causal estimators, people have created various benchmarks, each bringing different strengths and weaknesses that we will cover in Section 3. In this paper, we focus on how well causal estimators perform in the simplest setting, where there is no unobserved confounding, no selection bias, and no measurement error. It is straightforward to extend RealCause to these more complex settings. The ideal benchmark for choosing between causal estimators in this setting should have the following qualities: (1) yield ground-truth estimands, (2) be representative of a substantial subset of real data, (3) do not have unobserved confounders, and (4) yield many different data distributions of varying important characteristics (e.g. degree of overlap).

(1) is important in order to know which estimators yield estimates closer to the ground-truth. (2) is important so that we know that estimators that perform well on our benchmark will also perform well

Submitted to the 35th Conference on Neural Information Processing Systems (NeurIPS 2021) Track on Datasets and Benchmarks. Do not distribute.

on real datasets that we would apply them to. (3) is important so that we can rule out unobserved confounding as the explanation for an estimator performing poorly. (4) is important because it is unlikely that rankings of causal estimators on a single problem will generalize perfectly to all problems. Rather, we might expect that certain estimators perform better on distributions with certain properties and other estimators perform better on distributions with other specific properties. Existing benchmarks often have 1-3 of the above qualities (Section 3). Our benchmarking framework has all four.

We present a benchmark that simulates data from data generating processes (DGPs) that are statistically indistinguishable from observed real data. We first take the observed pretreatment covariates $W$ as the only common causes of $T$ and $Y$. Then, we fit generative models $P_{\text{model}}(T \mid W)$ and $P_{\text{model}}(Y \mid T, W)$ that closely match the real analogs $P(T \mid W)$ and $P(Y \mid T, W)$. This allows us to simulate realistic data by first sampling $W$ from the real data, then sampling $T$ from $P_{\text{model}}(T \mid W)$, and finally sampling $Y$ from $P_{\text{model}}(Y \mid T, W)$. Importantly, because we've fit generative models to the data, we can sample from *both* interventional distributions $P_{\text{model}}(Y \mid \text{do}(T = 1), W)$ and $P_{\text{model}}(Y \mid \text{do}(T = 0), W)$, which means that we have access to ground-truth estimands for our realistic simulated data. That is, the fundamental problem of causal inference isn't a problem in these DGPs. We then use this realistic simulated data for benchmarking.

**Main contributions**

1. RealCause and corresponding realistic benchmarks

2. Application of RealCause to show evidence in favor of selecting hyperparameters based on predictive metrics (like in machine learning)

3. Open-source dataset for predicting causal performance of causal estimators from predictive performance

## 2 Preliminaries and notation

Let $T$ be a binary scalar random variable denoting the treatment. Let $W$ be a set of random variables that corresponds to the observed covariates. Let $Y$ be a scalar random variable denoting the outcome of interest. Let $e(w)$ denote the *propensity score* $P(T = 1|W = w)$. We denote the treatment and outcome for unit $i$ as $T_i$ and $Y_i$. $Y_i(1)$ (resp. $Y_i(0)$) denotes the potential outcome that unit $i$ would observe if $T_i$ were 1, taking treatment (resp. if $T_i$ were 0, not taking treatment). $Y(t)$ is a random variable that is a function of all the relevant characteristics $I$ (a set of random variables) that characterize the outcome of an individual (unit) under treatment $t$.

We define the *individual treatment effect* (ITE) for unit $i$ as follows: $\tau_i \triangleq Y_i(1) - Y_i(0)$ We define the *average treatment effect* (ATE) as follows: $\tau \triangleq \mathbb{E}[Y(1) - Y(0)]$. Let $C$ be a set of random variables, denoting all the common causes (confounders) of the causal effect of $T$ on $Y$. We can identify the ATE from observational data if we observe $C$. This setting has many names: "no unobserved confounding," "conditional ignorability," "conditional exchangeability," 'selection on observables," etc. In this setting, we can identify the ATE via the *adjustment formula* (Robins, 1986; Spirtes et al., 1993; Pearl et al., 2016; Pearl, 2009):

$$\tau = \mathbb{E}_C \left[ \mathbb{E}[Y \mid T = 1, C] - \mathbb{E}[Y \mid T = 0, C] \right] \tag{1}$$

We define the *conditional average treatment effect* (CATE) similarly:

$$\tau(x) \triangleq \mathbb{E}[Y(1) - Y(0) \mid X = x] = \mathbb{E}_C \left[ \mathbb{E}[Y|T = 1, x, C] - \mathbb{E}[Y|T = 0, x, C] \right] \tag{2}$$

Here, $X$ is a set of random variables that corresponds to the characteristics that we are interested in measuring more specialized treatment effects with respect to ($x$-specific treatment effects). In this paper, we'll only consider CATEs where $X = W$, so there is no further need for the variable $X$.

Similarly, we consider DGPs where $W = C$, for simplicity, so it suffices to use only the variable $W$. This means that we must adjust for all of $W$ to get causal effects and that the CATEs reduce to

$$\tau(w) = \mathbb{E}[Y|T = 1, w] - \mathbb{E}[Y|T = 0, w] \triangleq \mu(1, w) - \mu(0, w), \tag{3}$$

where $\mu$ is the *mean conditional outcome*. Our DGPs provide ground-truth CATEs by providing $\mu$. This allows our DGPs to capture unobserved causes of $Y$ in the data.

## 3 Methods for evaluating causal estimators

### 3.1 Simulated synthetic data

The simplest way to get ground truth ATEs is to simulate synthetic data that we construct so that the only confounders of the effect of $T$ on $Y$ are $W$. This gives us access to the true *outcome mechanism* $P(Y \mid T, W)$. Using the outcome mechanism, we have access to the ground-truth CATE via Equation 3 and the ground-truth ATE via Equation 1.

In these simulations, we additionally have access to the true *treatment selection mechanism* $P(T \mid W)$ (or just "*selection mechanism*" for short). We must be able to sample from this to generate samples from $P(W, T, Y)$ via ancestral sampling: $P(W) \rightarrow P(T \mid W) \rightarrow P(Y \mid T, W)$. Having access to $P(T \mid W)$ gives us ground-truth for things like the propensity scores and the degree of positivity/overlap violations.

This is probably the most common method for evaluating causal estimators. However, it has several disadvantages. First, the data is completely synthetic, so we do not know if the rankings of estimators that we get will generalize to real data. Second, authors proposing new causal estimators are naturally interested in synthetic data with specific properties that their estimator was developed to perform well on. This means that different synthetic data used in different papers cannot be used for a fair comparison.

### 3.2 Simulated semi-synthetic data with real covariates

One natural improvement on the completely synthetic data described in Section 3.1 is to make it more realistic by taking the covariates $W$ from real data. This means that $P(W)$ is realistic. Then, one can proceed with generating samples through ancestral sampling by simulating $P(T \mid W)$ and $P(Y \mid T, W)$ as arbitrary stochastic functions. One of the main advantages of this is that these stochastic functions can be made to have any properties that its designers choose, such as degree of nonlinearity, positivity violation, treatment effect heterogeneity, etc. (Dorie et al., 2019). This is what many current benchmarks do (Dorie et al., 2019; Shimoni et al., 2018; Hahn et al., 2019). The main problem is that the selection mechanism $P(T \mid W)$ and outcome mechanism $P(Y \mid T, W)$ are unrealistic.

### 3.3 Simulated data that is fit to real data

The way to fix the unrealistic selection and outcome mechanisms is to fit them to real data. This is what we do, and we are not the first. For example, there is work on this in economics (Knaus et al., 2018; Athey et al., 2019; Huber et al., 2013; Lechner & Wunsch, 2013), in healthcare (Wendling et al., 2018; Franklin et al., 2014), and in papers that are meant for a general audience (Abadie & Imbens, 2011; Schuler et al., 2017). Some fit relatively simple models (Franklin et al., 2014; Abadie & Imbens, 2011), whereas others fit more flexible models (Wendling et al., 2018; Athey et al., 2019; Schuler et al., 2017). Our work is distinguished from the above work in two key ways: we statistically test that our generative models are realistic using two samples tests and we provide knobs to vary important characteristics of the DGPs. See Appendix A.1 for more discussion on this.

**Using RCTs for ground-truth** Finally, there are several different ways to use RCTs for ground-truths, but they all have problems, which we discuss in Appendix A.2.

## 4 RealCause: a method for producing realistic benchmark datasets

The basic idea is to fit flexible generative models $P_{\text{model}}(T \mid W)$ and $P_{\text{model}}(Y \mid T, W)$ to the selection mechanism $P(T \mid W)$ and the outcome mechanism $P(Y \mid T, W)$, respectively. For $P_{\text{model}}(W)$, we simply sample from $P(W)$, just as is done in the semi-synthetic data simulations we described in Section 3.2. These three mechanisms give us a joint $P_{\text{model}}(W, T, Y)$ that we would like to be the same as the true $P(W, T, Y)$. This is what makes our DGPs realistic.

**Architecture** We use neural networks to parameterize the conditioning of $P_{\text{model}}(T \mid W)$ and $P_{\text{model}}(Y \mid T, W)$; that is, the input of the neural net is either $W$ (to predict $T$) or both $W$ and $T$ (to predict $Y$). A naive approach would be to concatenate $W$ and $T$ to predict the $Y$, but our

experiments on semi-synthetic data (where $\tau$ is known) suggest that the resulting generative model tends to underestimate $\tau$. For example, this can happen from the network "ignoring" $T$, especially when $W$ is high-dimensional. Therefore, we follow the TARNet structure (Shalit et al., 2017) to learn two separate conditionals $P_{\text{model}}(Y \mid T = 0, W)$ and $P_{\text{model}}(Y \mid T = 1, W)$, encoding the importance of $T$ into the structure of our network. Since all conditionals depend on $W$, we use a multi-layer perceptron (MLP) to extract common features $h(W)$ of $W$. We then have three more MLPs to model $T$, $Y \mid T = 0$, and $Y \mid T = 1$ separately, taking in the features $h(W)$ as input. These all use the same $h(W)$, which is also learned, like in Dragonnet (Shi et al., 2019). For simplicity, all four MLPs have the same architecture. The tunable hyperparameters are the number of layers, the number of hidden units, and the activation function.

**Distribution assumption**    We use the output of the MLPs to parameterize the distributions of selection and outcome. For example, for binary data (such as treatment), we apply the logistic sigmoid activation function to the last layer to parameterize the mean parameter of the Bernoulli distribution. For real-valued data (such as the outcome variable), one option is to assume it follows a Gaussian distribution conditioned on the covariates, in which case we would have the neural net output the mean and log-variance parameters. The baseline model that we use is a linear model that outputs the parameters of a Gaussian distribution with a diagonal covariance matrix. The main (more flexible) generative model we use is the sigmoidal flow (Huang et al., 2018), which has been shown to be a universal density model capable of fitting arbitrary distributions.

For mixed random variables, we parameterize the likelihood as a mixture distribution: $P(Y) = \pi_0 1_{Y \notin \mathcal{A}} P_c(Y) + \sum_{j=1}^{K} \pi_j 1_{Y = a_j}$ where $\mathcal{A} = \{a_1, ..., a_K\}$ is the set of (discrete) atoms, $\pi_j$ for $j = 0, ..., K$ forms a convex sum, and $P_c$ is the density function of the continuous component. We have dropped the conditioning to simplify the notation.

**Optimization**    For all the datasets, we use a 50/10/40 split for the training set, validation set, and test set. To preprocess the covariate ($W$) and the outcome ($Y$), we either standardize the data to have zero mean and unit variance or normalize it so that the training data ranges from 0 to 1. We use the Adam optimizer to maximize the likelihood of the training data, and save the model with the best validation likelihood for evaluation and model selection. We perform grid search on the hyperparameters and select the model with the best (early-stopped) validation likelihood and with a p-value passing 0.05 on the validation set.

**Tunable knobs**    After we fit a generative model to a dataset, we might like to get other models that are very similar but differ along important dimensions of interest. For example, this will allow us to test estimators in settings where there are positivity/overlap violations, where the causal effect is large/small, or where there is a lot of heterogeneity, no heterogeneity, etc. To do this, RealCause supports the following 3 knobs that we can turn to generate new but related distributions, after we've fit a model to a real dataset.

*Positivity/overlap knob*    Let $p_i$ be the probability of treatment for example $i$ (i.e. $p_i = P(T = 1 \mid W = w_i)$). The value of this knob $\beta$ can be set to anywhere between 0 and 1 inclusive. We use $\beta$ to linearly interpolate between $p_i$ and the the extreme that $p_i$ is closer to (0 or 1). Namely, we change $p_i$ to $p_i'$ according to the following equation: $p_i' = \beta p_i + (1 - \beta) 1_{p_i \geq 0.5}$. For example, $\beta = 1$ corresponds to the regular data, $\beta = 0$ corresponds to the setting where treatment selection is fully deterministic, and all other values of $0 < \beta < 1$ correspond to somewhere in between.

*Heterogeneity knob*    The value $\gamma$ of the heterogeneity knob can be any real value between 0 and 1 inclusive. If $\gamma$ is set to 1, the CATEs are the same as the regular dataset. If $\gamma$ is set to 0, the CATEs are all equal to the ATE. If $\gamma$ is somewhere between 0 and 1, the CATEs are the corresponding linear interpolation of the original CATE and the ATE.

*Causal effect scale knob*    The value $s$ of the causal effect scale knob can be any real number. This knob sets the scale of the causal effects by changing the potential outcomes according to the following equations: $Y_i(1)' = s \frac{Y_i(1)}{\tau}$ and $Y_i(0)' = s \frac{Y_i(0)}{\tau}$.

# 5    How realistic is RealCause?

In this section, we show that RealCause produces realistic datasets that are very close to the real ones. For all datasets, we show that the distribution of our generative model $P_{\text{model}}(W, T, Y)$ is

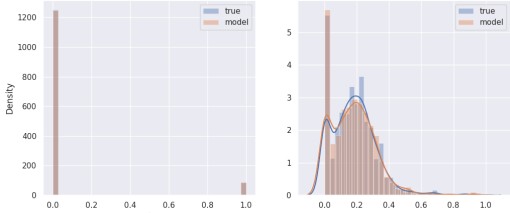 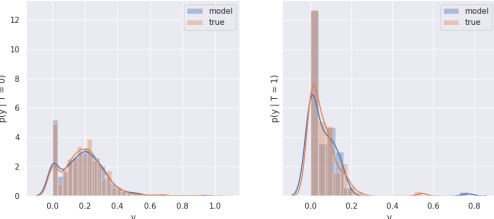

(a) Marginal distributions $P(T)$ and $P_{\text{model}}(T)$ on the left and marginal distributions $P(Y)$ and $P_{\text{model}}(Y)$ on the right.

(b) Histogram and kernel density estimate visualization of $P(Y \mid T)$ and $P_{\text{model}}(Y \mid T)$, sharing the same y-axis.

Figure 1: Visualizations of how well the generative model models the real LaLonde PSID data.

very close to the true distribution $P(W, T, Y)$. We show this by providing both visual comparisons and quantitative evaluations. We visually compare $P_{\text{model}}(T, Y)$ and $P(T, Y)$ using histograms and Gaussian kernel density estimation (see, e.g., Figure 1). We quantitatively compare $P_{\text{model}}(W, T, Y)$ and $P(W, T, Y)$ by running two-sample tests (Table 1).

Two-sample tests evaluate the probability that a sample from $P_{\text{model}}(W, T, Y)$ and a sample from $P(W, T, Y)$ came from the same distribution, under the null hypothesis that $P_{\text{model}}(W, T, Y) = P(W, T, Y)$ (that the model distribution matches the true distribution). If that probability (p-value) is less than some small value $\alpha$ such as $0.05$, we say we have sufficient evidence to reject the null hypothesis that $P_{\text{model}}(W, T, Y) = P(W, T, Y)$ (i.e. the generative model is not as realistic as we would like it to be). This is how we operationalize the hypothesis that our modeled distributions are "realistic." Two-sample tests give us a way to falsify the hypothesis that our generative models are realistic.

However, two-sample tests do not work well in high-dimensions. Importantly, the power[1] of two-sample tests can decay with dimensionality (Ramdas et al., 2015) and $W$ can have many dimensions in the datasets we consider. On the bright side, the treatment $T$ and the outcome $Y$ are each one-dimensional, so evaluating the statistical relationship between them is only a two-dimensional problem. This means that we might get more power from testing the hypothesis that $P_{\text{model}}(T, Y) = P(T, Y)$ because it's a lower-dimensional problem, even though this test will ignore $W$ and its relationship to $T$ and $Y$. Tests that use $P(W, T, Y)$ could have more power because they use information about $P(T, Y \mid W)$ (recall that $P(W) = P_{\text{model}}(W)$, by construction). Therefore, we run two-sample tests for both $P(T, Y)$ and $P(W, T, Y)$ (and the marginals). Finally, we stress that passing the marginal tests is not trivial, since we learn the conditional $P(T, Y \mid W)$ and marginalize out $P(W)$, instead of learning $P(T, Y)$, $P(T)$, or $P(Y)$ directly.

**Datasets** We fit eight datasets in total. We fit generative models to three real datasets: LaLonde PSID, LaLonde CPS (LaLonde, 1986) (we use Dehejia & Wahba (1999)'s version), and Twins[2] (Louizos et al., 2017). We additionally fit generative models to five popular semi-synthetic datasets: IHDP (Hill, 2011) and four LBIDD datasets (Shimoni et al., 2018). On all of these datasets, we can fit generative models to model the observational distribution. Then, with the semi-synthetic datasets, we can also check that our generative models give roughly the same ground-truth causal effects as existing popular synthetic benchmarks.

**Visualization of modeled LaLonde PSID** Consider the LaLonde PSID dataset as our first example. We visualize $P_{\text{model}}(T)$ vs. $P(T)$ and $P_{\text{model}}(Y)$ vs. $P(Y)$ in Figure 1a. $P_{\text{model}}(W)$ and $P(W)$ are known to be the same distributions, by construction. We visualize $P_{\text{model}}(T, Y)$ vs. $P(T, Y)$ in Figure 1b. We provide similar visualizations of the other real datasets and corresponding similar models in Appendix B.

**Univariate statistical tests** The Kolmogorov-Smirnov (KS) test is the most popular way to test the hypothesis that two samples come from the same distribution. The Epps-Singleton (ES) test is more well-suited for discrete distributions and can have higher power than the KS test (Epps &

---

[1]For a fixed value of $\alpha$, *power* is the probability of rejecting the null hypothesis, given that the null hypothesis is false.

[2]The treatment selection mechanism for the Twins dataset is simulated. This is to ensure that there is some confounding, as the regular dataset might be unconfounded.

Singleton, 1986). We use the implementations of the KS and ES tests from *SciPy* (Virtanen et al., 2020). For all datasets, we report the p-values of the KS and ES tests for comparing the marginal distributions $P_{\text{model}}(Y)$ and $P(Y)$ and for comparing the marginal distributions $P_{\text{model}}(T)$ and $P(T)$ in the first section of Table 1. In all tests, the p-values are much larger than any reasonable value of $\alpha$, so we fail to reject the null hypothesis that the generated data and the true data come from the same distribution. This means that our generative models are reasonably realistic, at least if we only look at the marginals.

**Multivariate statistical tests** Extending the KS test to multiple dimensions is difficult. However, there are several multivariate tests such as the Friedman-Rafsky test (Friedman & Rafsky, 1979), k-nearest neighbor (kNN) test (Friedman & Rafsky, 1983), and energy test (Székely & Rizzo, 2013). We use the implementations of these tests in the *torch-two-sample* Python library (Djolonga, 2017). These are just permutation tests and can be conducted with any statistic, so we additionally run permutation tests with the Wasserstein-1 and Wasserstein-2 distance metrics. We run each test with 1000 permutations. We display the corresponding p-values in the last two sections of Table 1. For all tests except the FR and kNN $(T, Y)$ test on the LaLonde PSID dataset, the p-values are much larger than any reasonable value of $\alpha$. However, we might be worried that these multivariate two-sample tests don't have enough power when we include the higher-dimensional $W$.

**Demonstration of statistical power via linear baselines** We demonstrate that these tests do have a decent amount of statistical power (probability of rejecting the null when $P_{\text{model}}$ and $P$ differ) by fitting a linear Gaussian model to the data and displaying the corresponding p-values in Table 2. Even when $W$ is high-dimensional, we are still able to reject the linear models as realistic. For example, we clearly have p-values that are below most reasonable values of $\alpha$ for the LaLonde PSID, and all three nonlinear LBIDD datasets. As we might expect, for high-dimensional $W$ such as in the LBIDD datasets, the $(T, Y)$ tests have enough power to reject the null hypothesis because they operate in only two dimensions, whereas the $(W, T, Y)$ tests do not because their power suffers from the high-dimensionality (179 dimensions). The LaLonde CPS dataset is an example where it can be useful to include $W$ in the statistical test; all of the p-values for the $(T, Y)$ tests are *above* $\alpha = .075$, whereas all but one of the p-values for the $(W, T, Y)$ tests are *below* $\alpha = .075$. Our p-values for the Twins dataset are quite high, but this is not due to these tests not having enough power. Rather, it is because the Twins dataset is well modeled by a linear model: $T$ and $Y$ are both binary (two parameters) and $W$ is 75-dimensional, so it makes sense that we can linearly predict these two parameters from 75 dimensions. We demonstrate how well the linear model fits Twins in

Table 1: Table of p-values for the various statistical hypothesis tests we run to test the null hypothesis that real data samples and samples from the generative model come from the same distribution. Large values (e.g. > 0.05) mean that we don't have statistically significant evidence that the real and generated data come from different distributions, so we want to see large values. The first section is univariate tests. The second section is 2-dimensional tests to capture the dependence of $Y$ on $T$. The third section can be much higher dimensional tests whose power may suffer from the high dimensionality, but these tests may be able to pick up on the dependence of $T$ and $Y$ on $W$ that the 2-dimensional tests cannot pick up on.

| | LALONDE | | | | LBIDD | | | |
|---|---|---|---|---|---|---|---|---|
| TEST | PSID | CPS | TWINS | IHDP | QUAD | EXP | LOG | LINEAR |
| $T$ KS | 0.9995 | 1.0 | 0.9837 | 0.9290 | 0.5935 | 0.9772 | 0.4781 | 0.3912 |
| $T$ ES | 0.6971 | 0.3325 | 0.7576 | 0.5587 | 0.8772 | 0.6975 | 0.4157 | 0.3815 |
| $Y$ KS | 0.4968 | 1.0 | 0.8914 | 0.3058 | 0.2204 | 0.9146 | 0.4855 | 0.4084 |
| $Y$ ES | 0.3069 | 0.1516 | 0.4466 | 0.3565 | 0.2264 | 0.7223 | 0.3971 | 0.1649 |
| $(T, Y)$ Wass1 | 0.6914 | 0.435 | 0.5088 | 0.2894 | 0.3617 | 0.4391 | 0.3899 | 0.5046 |
| $(T, Y)$ Wass2 | 0.6638 | 0.4356 | 0.4960 | 0.3365 | 0.4353 | 0.4709 | 0.4205 | 0.5063 |
| $(T, Y)$ FR | 0.0 | 0.4004 | 0.5549 | 0.4761 | 0.8610 | 0.5773 | 0.5132 | 0.8355 |
| $(T, Y)$ kNN | 0.0 | 0.4120 | 0.4318 | 0.5978 | 0.3166 | 0.3735 | 0.4902 | 0.4838 |
| $(T, Y)$ Energy | 0.6311 | 0.4396 | 0.5053 | 0.3186 | 0.2371 | 0.4453 | 0.3988 | 0.5086 |
| $(W, T, Y)$ Wass1 | 0.4210 | 0.3854 | 0.4782 | 1.0 | 0.5191 | 0.4219 | 0.4866 | 0.5393 |
| $(W, T, Y)$ Wass2 | 0.5347 | 0.3660 | 0.4728 | 1.0 | 0.5182 | 0.4160 | 0.4807 | 0.5381 |
| $(W, T, Y)$ FR | 0.2569 | 0.4033 | 0.5068 | 1.0 | 0.4829 | 0.4989 | 0.5027 | 0.4893 |
| $(W, T, Y)$ kNN | 0.2270 | 0.4343 | 0.4919 | 1.0 | 0.5104 | 0.5101 | 0.5223 | 0.4988 |
| $(W, T, Y)$ Energy | 0.5671 | 0.4177 | 0.5263 | 0.9409 | 0.5104 | 0.4423 | 0.5031 | 0.5421 |
| $|W|$ (n covariates) | 8 | 8 | 75 | 25 | 177 | 177 | 177 | 177 |

Table 2: Table of p-values for the various statistical hypothesis tests we run to test the null hypothesis that real data samples and samples from a *linear* Gaussian generative model come from the same distribution. Small values (e.g. < 0.05) mean that these tests have enough power to detect that the real data comes from a different distribution than the distribution generated by our linear Gaussian generative model.

| | LALONDE | | | | LBIDD | | | |
| TEST | PSID | CPS | TWINS | IHDP | QUAD | EXP | LOG | LINEAR |
|---|---|---|---|---|---|---|---|---|
| $(T, Y)$ Wass1 | 0.0304 | 0.1500 | 0.5004 | 0.2019 | 0.2009 | 0.0456 | 0.1510 | 0.2832 |
| $(T, Y)$ Wass2 | 0.0123 | 0.0797 | 0.4924 | 0.1636 | 0.4277 | 0.1314 | 0.2380 | 0.3172 |
| $(T, Y)$ FR | 0.0 | 0.0776 | 0.5581 | 0.2825 | 0.0 | 0.0014 | 0.0140 | 0.7946 |
| $(T, Y)$ kNN | 0.0 | 0.1808 | 0.4541 | 0.4183 | 0.0 | 0.0023 | 0.0013 | 0.4070 |
| $(T, Y)$ Energy | 0.0482 | 0.1620 | 0.5094 | 0.2249 | 0.0002 | 0.0551 | 0.2020 | 0.3409 |
| $(W, T, Y)$ Wass1 | 0.0470 | 0.0671 | 1.0 | 1.0 | 0.4917 | 0.5245 | 0.8230 | 0.6777 |
| $(W, T, Y)$ Wass2 | 0.4001 | 0.0624 | 0.9966 | 1.0 | 0.4782 | 0.5204 | 0.7840 | 0.6257 |
| $(W, T, Y)$ FR | 0.1333 | 0.0525 | 0.9992 | 1.0 | 0.7655 | 0.6979 | 0.3651 | 0.7369 |
| $(W, T, Y)$ kNN | 0.5136 | 0.0711 | 1.0 | 1.0 | 0.8953 | 0.8416 | 0.4510 | 0.7968 |
| $(W, T, Y)$ Energy | 0.1080 | 0.2863 | 0.7389 | 0.8935 | 0.5099 | 0.5142 | 0.7429 | 0.7144 |
| $|W|$ (n covariates) | 8 | 8 | 75 | 25 | 177 | 177 | 177 | 177 |

Figures 5c and 5d in Appendix B. Similarly, the p-values for IHDP are so high because the IHDP data is reasonably well fit by the linear model (see Figures 6d to 6f), and the IHDP tests have less power since the IHDP dataset is much smaller than the other datasets.

**Realistic causal effects**  We also show that our generative model admits causal effect estimates that roughly match those of the popular semi-synthetic benchmarks IHDP and LBIDD. For each of these datasets, we report the true ATE, our generative model's ATE estimate, the corresponding absolute bias, and the PEHE. We report these values in Table 3. The values in the table indicate that our model accurately models the causal effects. The one number that is relatively high relative to the others is the PEHE for IHDP; this is because the training sample for IHDP is only 374 examples.

**Limitations**  Although we can statistically test how well RealCause fits the observed distribution $P(W, T, Y)$, we cannot test how well RealCause fits the interventional distributions $P(Y \mid \mathrm{do}(T = t), w)$ without making the no unobserved confounding assumption. Due to the fundamental problem of causal inference, there is no way of getting around this for arbitrary distributions. Fortunately, we can test the interventional distributions of synthetic data such as IHDP and LBIDD; this is why we include Table 3. That said, RealCause (or any realistic benchmark) could potentially not model the interventional distributions well on other datasets, resulting in suboptimal interventional distributions. Additionally, RealCause will be biased based on the specific architecture of the generative model it uses. Ideally, one would run RealCause benchmarks using many different generative model architectures.

# 6   Results

The reason we spent so much effort establishing that RealCause DGPs are realistic in Section 5 is that we can now trust the results that RealCause DGPs yield for important tasks such as the following: (a) benchmarking causal estimators and (b) evaluating whether *predictive* metrics can be used for model selection of *causal* estimators. We first apply RealCause to benchmarking causal estimators. We then use these results to analyze correlation between predictive performance and causal performance in Section 6.1.

Table 3: True causal effects, corresponding estimates from our generative model, and associated error.

| | IHDP | LBIDD QUAD | LBIDD EXP | LBIDD LOG | LBIDD LINEAR |
|---|---|---|---|---|---|
| True ATE | 4.0161 | 2.5437 | -0.6613 | 0.0549 | 1.8592 |
| ATE estimate | 4.1908 | 2.4910 | -0.6608 | 0.0555 | 1.7177 |
| ATE abs bias | 0.1747 | 0.0527 | 0.0004 | 0.0005 | 0.1415 |
| PEHE | 51.5279 | 0.1554 | 0.0225 | 0.0151 | 0.1367 |

**Datasets and estimators**   In our evaluations in this section, we use 3 real datasets, 4 meta-estimators, 15 machine learning models for each of the meta-estimators, and roughly 10 different settings of the single most important hyperparameter for each of the machine learning models. Taking the Cartesian product over all of those yields over 1500 causal estimators. The 3 datasets we use are LaLonde PSID, LaLonde CPS, and Twins; we use RealCause to turn these into datasets where we know the ground-truth causal effects. The 4 meta-estimators from *causallib* (Shimoni et al., 2019) we use are standardization (or S-learner), stratified standardization (or T-learner), inverse probability weighting (IPW), and IPW with weight trimming. We use a variety of machine learning models from *scikit-learn* (Pedregosa et al., 2011) to plug in to these meta-estimators. For each model, we use a grid of values for the most important hyperparameter (according to van Rijn & Hutter (2018)). See Appendix E for more info on our estimators.

**Benchmarking causal estimators**   As one would expect, different causal estimators perform better on different datasets. We choose causal estimators within a given model class according to the best cross-validated RMSE for standardization estimators and according to the best cross-validated average precision for IPW estimators. We divide the ATE RMSEs by each dataset's ATE and show those weighted averages in Figure 2. Interestingly, most of our standardization estimators don't perform very well, but then standardization pair with an RBF-SVM achieves the lowest ATE RMSE. While this estimator also achieves the lowest weighted averaged PEHE, it doesn't have the lowest weighted averaged absolute bias. We provide the corresponding plots for ATE absolute bias and PEHE along with the more fine-grained full tables by dataset in Appendix C.

### 6.1   Predicting causal performance from predictive performance

The following is known and commonly stated: just because the model(s) used in a causal estimator are highly predictive does not mean that the causal estimator will perform well at estimating a causal parameter such as $\tau$ or $\tau(w)$. Then, the following questions naturally arise: (1) How can I choose hyperparameters for causal estimators? (2) How can I inform model selection for causal problems? In machine learning, the answer is simple: run cross-validation using the relevant predictive metric for hyperparameter and model selection. However, we can't do the analog in causal inference because we don't have access to a corresponding causal metric, due to the fundamental problem of causal inference.

What if it turns out that the hyperparameters and models that yield the best predictive performance also yield the best causal performance? Then, hyperparameter and model selection for causal inference would be the same as it is for machine learning. We can measure if this is the case by measuring how correlated predictive metrics and causal metrics are.

**Correlation measures**   While Pearson's correlation coefficient is the most common method for measuring correlation, it only captures linear relationships. We are more interested in general monotonic relationships (e.g. if the prediction performance of model A is better than the predictive performance of model B, then will the causal performance of model A also be better than the causal performance of model B?). Therefore, we use Spearman's rank correlation coefficient (equivalent to Pearson's correlation coefficient on the *rank* of the random variables) and Kendall's rank correlation coefficient. We also report a more intuitive measure: the probability that the causal performance of model A is at least as good as the causal performance of model B, given that the predictive performance of model A is at least as good as the predictive performance of model B.

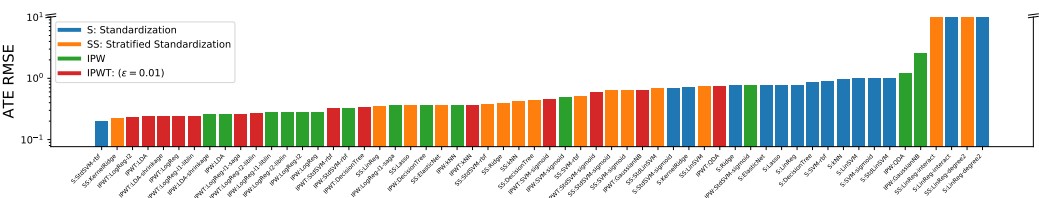

Figure 2: ATE RMSE of the different estimators, weighted averaged (by their inverse ATEs) over three datasets and color-coded by meta-estimator.

**Metrics** The main predictive metric we consider for outcome models (predict $Y$ from $T$ and $W$) is the corresponding RMSE (root mean squared error). The main predictive metrics we consider for propensity score models (predict binary $T$ from $W$) are scikit-learn's balanced F score, average precision, and balanced accuracy. The main causal metric is the PEHE (Hill, 2011).

**Selecting model hyperparameters** For a given dataset, meta-estimator, and machine learning model class, we must choose the hyperparameters for that specific model class. We show the full table of correlation coefficients for how predictive RMSE is of ATE RMSE and PEHE within every model class in Appendix D.1. We summarize this with just the median Spearman correlation coefficient and the median probability of better or equal causal performance given better or equal predictive performance in Table 4; these medians are taken over all models for standardization and stratified standardization estimators fit to a given dataset. **Importantly, these results show that, in this setting, it is a fairly good idea to select hyperparameters for causal estimators based on predictive performance.** For example, the median probabilities that a better predictive model corresponds to a better causal model hover around 80-95% in this summary table. We do the same for IPW and propensity score models in Appendix D.2.

Table 4: Median correlation of predictive RMSE with PEHE in standardization estimators.

| DATASET | SPEARMAN | PROB BETTER |
|---------|----------|-------------|
| PSID | 0.92 | 0.92 |
| CPS | 0.80 | 0.87 |
| Twins | 0.91 | 0.96 |

**Model selection** We just saw that predictive performance is indicative of causal performance when choosing hyperparameters within a model class, but what about selecting between model classes after choosing hyperparameters via predictive cross-validation? The results are much less positive and more dataset-specific. For standardization estimators, there isn't much correlation on the LaLonde datasets, but there is a great deal of correlation on the Twins dataset. For IPW estimators, it is roughly the same, except for the fact that average precision has a modest correlation with ATE RMSE on the LaLonde CPS dataset. See Appendix D.3 for details.

**Open-source dataset for exploration** We created a dataset with 1568 rows (estimators) and 77 columns (predictive metrics, causal metrics, and estimator specification). Importantly, this dataset contains all the predictive metrics that scikit-learn provides and many different causal metrics that we compute using RealCause. In this section, we chose one line of analysis for this dataset, but there are many others. For example, one can use any machine learning model for predicting any subset of causal metrics from any subset of predictive metrics, one can cross-validate over different predictive metrics than the ones we used, one can group the data differently, etc. We already see that different predictive metrics correlate quite differently with ATE RMSE, depending on the model and dataset in Appendix D.2. This suggests that more value might be gained in doing more complex analyses on this dataset. We open-source our dataset at `https://github.com/bradyneal/causal-benchmark/blob/master/causal-predictive-analysis.csv`.

## 7 Conclusion and future work

Now that we've rigorously shown that RealCause produces realistic DGPs, we are hopeful that others will use it. We open-source default benchmark datasets, our trained RealCause generative models, and the code to train new generative models on other datasets at `https://github.com/bradyneal/causal-benchmark`.

There are many important extensions of RealCause that can be done. Adding even more causal estimators and more real datasets would be valuable to expand the open-source dataset of predictive and causal metrics that we started. Similarly, running the benchmarking suite with various non-default settings of RealCause's knobs (e.g. zero overlap) could lead to useful empirical results about when to use various estimators. RealCause's realism gives us confidence in our evidence that hyperparameters for causal estimators can be selected using cross-validation on a predictive metric. There is much potential for further analysis of our open-source dataset of predictive and causal metrics. For example, future papers or a Kaggle competition to predict causal metrics from predictive metrics would be valuable.

## Acknowledgements

We thank Uri Shalit, Yoshua Bengio, and Ioannis Mitliagkas for useful feedback on this paper.

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
