# OpenReview forum: "RealCause: Realistic Causal Inference Benchmarking"
_NeurIPS.cc/2021/Track/Datasets_and_Benchmarks/Round1 — Submitted to NeurIPS 2021 Datasets and Benchmarks Track (Round 1)_

### Official Review · Reviewer_rWPg · 2021-07-02
**Proposes framework for generating causal inference datasets using generative models**

**Rating:** 6
**Confidence:** 4

**Strengths:**

The paper does a great job of motivating the importance of the problem at hand: realistic datasets that contain ground-truth potential outcomes. This problem and the proposed solutions are very relevant for researchers in causal inference. The limitations section on line 264 was well-done and the authors should be commended for explicitly noting limitations.



**Weaknesses:**

The significance of the overall contribution is tenuous. Among the contributions, the most compelling was the least novel: a generative approach for causal inference datasets. Although not novel, this part was well-written and well-motivated.

I have several questions/concerns about the more novel contributions (the two-sample tests, tunable knobs, analysis on predictive performance predicting causal performance). Regarding the use of two-sample tests to test the hypothesis that the generated joint distribution of observables is different from the actual observed joint distribution, a notable issue is that in this setup we care about type II error because we care about mistaking our generated dataset as realistic when it is not. But hypothesis tests are designed to control type I error, so failing to reject the null is not compelling evidence. The paper begins to address this with an analysis of power using linear baselines, but this is very limited and the results are not guaranteed to hold for other settings.

I found the proposal on the tunable knobs for varying overlap, causal effect strength, and heterogeneity to be interesting but under-developed. It would be helpful to motivate these choices for tuning knobs. Could there exist a more optimal method that varies a dimension (like overlap) while minimizing the deviation from the original distribution? And it appears these knobs were not used in the reported experiments. Was there rationale for that?

I found the section on using predictive performance to predict causal performance to be confusing. What does "predictive performance" refer to? This section discusses hyper-parameter tuning and model selection for "causal estimators" but it's not clear whether that refers to the nuisance function estimation or the meta-estimator or both?

Finally, it would be helpful to elaborate on how estimates like P_model(T) in Figure 1 and ATE estimate in table 3 are computed. Are these based on the test set? And it is possible to give standard errors?


**Additional Feedback:**

No additional feedback

**Clarity:**

The paper is fairly well written.

The sections that would benefit from revision are :
- 6.1. As noted above, 6.1 is confusing.
- It would also be helpful to clarify the interpretation of the original dataset: Does this paper treat the data as a finite population or as samples from the target? For instance the paper refers to the covariate distribution P(W) (line 204) but it's not clear whether that is estimated or observed (if the data is taken to be the finite population).

**Correctness:**

Noted above are a few concerns with correctness in the following claim:
- the proposed two-sample tests "rigorously" (line 357) show the framework proposes realistic DGPs.


**Documentation:**

There isn't sufficient detail in the main paper to support reproducibility for reasons noted above (how the probabilities and ATEs are computed). If these are provided in the appendix, it would be helpful to provide the reference.

**Ethics:**

No ethical concerns

**Relation To Prior Work:**

Strengths:
Line 119 provides a nice differentiation from prior work, noting that their method using two-sample tests and provides knobs to vary dimensions.

Weakness:
It would be helpful to provide more comparison to prior work in the datasets section on line 208 and to clarify the motivation here. Are the proposed generated datasets superior to the existing LaLonde,  IHDP, etc datasets?

**Summary And Contributions:**

This paper proposes a framework for creating datasets to evaluate causal effect estimators. The framework builds generative models for the treatment assignment mechanism and for the potential outcome models and then generates a sample as follows: sample from the observed covariate distribution, sample from the modeled treatment, and sample the corresponding potential outcome. The framework assesses the fit of this dataset by using a two-sample test to test the hypothesis that the generated joint distribution of observables (treatment, outcome, covariates) is different from the actual observed joint distribution of observables. The paper applies this framework to several commonly used causal inference datasets, including the Lalonde jobs dataset, IHDP, LBIDD, and twins. They use the generated datasets to benchmark 4 meta-estimators: S-learner, T-learner, IPW and IPW with weight clipping. They benchmark these methods over different learning methods and different hyperparameter settings. Finally the authors perform analysis of how well predictive performance predicts accurate causal effect estimation; however the meaning of "predictive performance" was not clear and this section in general was hard to follow.

---

> ### Author Response · Authors · 2021-07-11
> **Response**
>
> “Predictive performance” - Thanks for the feedback on the lack of clarity there; we’ll definitely clear this up in the paper. We mean the losses that evaluated how well the outcome and propensity score models are doing at predicting the outcome and treatment, respectively. An alternative dichotomy that we’ve recently considered is “factual” vs. “counterfactual” scores. All the “predictive” scores that we considered are “factual” in the sense that they only have access to factual (observed) data and the “causal” scores are “counterfactual” in the sense that they have access to the counterfactuals. This dichotomy is important because, in practice, we only have access to the factual (predictive) scores, but we care about the counterfactual (causal) scores, so it’s useful to know when the two are associated and when they aren’t.
>
> Two sample tests and type II error - We agree that the fact that the type II error is surely non-zero means that the model distributions and the true distributions could very well not be exact matches. But exact matches are not what we’re looking for. Rather, imagine you have a pool of many different generative models fit to the observed data and you have to narrow down to the ones that are sufficiently realistic. Many of the generative models that people fit to real data (e.g. in Section 3.3) would be rejected by the two sample tests that we run. In fact, it took a large amount of effort to even get our highly flexible models to not be rejected by these two sample tests. In other words, it is highly non-trivial to “pass” these tests (not reject the null). Going back to the many different generative models that you could imagine, the fact that these tests are not easy to “pass” means that we can remove many of these generative models from the pool, leaving only a small number of most realistic models (analogous to rejecting more and more hypotheses in the scientific method).
>
> Tunable knobs - We did not perform any optimality analysis of our choice of implementation for these knobs. This is primarily because we were most focused on just being able to implement these knobs in the generative model architecture. I think this would be a good direction and focus for future research / improvement of RealCause.

---

### Official Review · Reviewer_G4rW · 2021-07-05
**RealCause: Realistic Causal Inference Benchmarking**

**Rating:** 6
**Confidence:** 3
**Correctness:** The method is constructed in a sound …

**Strengths:**

1. Proposes a framework to generate benchmarks for causal inference, which facilitates quantitative evaluation in this field;
2. Provides the generated benchmark to evaluate existing causal inference models;
3. Code available to the public on GitHub.

**Weaknesses:**

1. This work can be better motivated. Specifically, what can be some applications of causal inference, and how this benchmark can help for these applications?
2. The advantage of this benchmark is not clear enough over prior methods for evaluating causal estimators. Specifically, it would be helpful to evaluate causal estimators using a prior estimation method and make a claim why the proposed benchmark is better.

**Additional Feedback:**

Please see above.

**Clarity:**

The paper is well-written and easy to follow. Minor points:
1. Fig 1(a) is hard to read.
2. It would be great for the authors to clarify the significance of the correlation between predictive and causal metrics.

**Documentation:**

Overall, a lot of details of the dataset are discussed. However, it would be great if the authors could add the points below:
1. The implementation details of the DGP and machine learning models need to be discussed in detail. Currently, they are not presented in the paper. Would the fitted distribution/causal inference performance be different if the model capacity is changed?
2. The process of adding a new dataset into the benchmark can be discussed in detail, specifically how to fit the models to the real data.

**Relation To Prior Work:**

Prior work is well-reviewed, but just like mentioned earlier, it would be great if the author can evaluate causal estimators on prior benchmarks and point out the advantage of the proposed work by making a comparison.

**Summary And Contributions:**

In this work, the authors proposed a framework to produce a benchmark to test causal estimators with data that claimed to be realistic. The framework makes use of real data and uses learned neural networks as the data generating processes to produce the data. The paper also mentions three "turnable knobs" which can be used to configure the output data distribution. The authors present the comparison between the real data and the generated data by showing statistical tests and visualizations of distributions. For the results, the authors ran 1500 configurations of datasets, meta-estimators, and machine learning models. This work also shows that predictive performance is a good indicator of the performance of causal estimation to select hyperparameters, but the model selection can be more dataset-specific.

---

> ### Author Response · Authors · 2021-07-11
> **Response**
>
> “advantage of this benchmark” -
> The main advantage of this benchmark over others is that we test how realistic ours is. Most of the ones referenced in Section 3.3 would not “pass” the many tests we run in Section 5. We know this because many of those generative models are not very flexible, and it took a large amount of effort to get our highly flexible models to not be rejected by these two sample tests.
>
> “It would be great for the authors to clarify the significance of the correlation between predictive and causal metrics.” -
> Great point. We will definitely work on the clarity there. Predictive metrics are meant to be metrics that only have access to observed (factual) data whereas causal metrics additionally have access to unobserved (counterfactual) data.

---

### Official Review · Reviewer_TQVn · 2021-07-05
**The paper presents a benchmarking framework for the realistic estimation of causal effects. The evaluation process in this work seems to be thorough but the overall importance of the contribution is unclear.**

**Rating:** 5
**Confidence:** 2

**Strengths:**

Causal inference is an active area of research which increasingly attracts the interest of ML researchers. In that sense, a causal inference benchmarking framework is timely and relevant to a broad ML audience.

**Weaknesses:**

My main concern with this paper is that, in my opinion, it lucks a clear narrative and contextualization. After reading the paper, it is still unclear to me how the authors measure the effectiveness of their benchmarking approach and its importance compared to prior work. In lines 113-121 the authors refer to a number of related papers which seem to do more or less what they do since they write “The way to fix the unrealistic selection and outcome mechanisms is to fit them to real data. This is what we do, and we are not the first.”. Then, they refer to two advantages of their benchmarking procedure by saying: “we statistically test that our generative models are realistic using two samples tests” and “we provide knobs to vary important characteristics of the DGPs”. However, it is unclear to me how important these two points are. The existence of the 3 knobs does not seem to be sufficiently discussed/experimented with. Moreover, the authors spend almost 3 pages in Section 6 to show that their proposed generative models do fit the real data. Perhaps I missed something important in the paper but I would like the authors to clarify the following: Is just fitting the datasets good enough to claim that this generative process provides a realistic causal model? Also, it is hard to evaluate whether this is true since there doesn’t seem to be a discussion about what these datasets actually are.

In general, it would be helpful if the authors clarified what is the main point that they are trying to make in each of the sections 5,6. To my reading, the text seemed a bit wordy and not particularly convincing about the need/usefulness of the proposed benchmarking framework.

--- After author response ---

I read the response and I would like to keep my current score (5). Although the authors shed some light into the aforementioned points with their response, I think those points are major and they weren't properly communicated in the submitted version of the paper. Therefore, in my opinion, the paper would need a revision to get ready for publication and I am hesitant to suggest its acceptance in the current form.

**Additional Feedback:**

For suggestions for improvement, look at the “Weaknesses” and “Clarity” sections.

Other than what I already mentioned, some extra questions for the authors are:
- Line 30: The authors write “In this paper, we focus on how well causal estimators perform in the simplest setting, where there is no unobserved confounding, no selection bias, and no measurement error. It is straightforward to extend RealCause to these more complex settings.”. It would be helpful to clarify how this is straightforward and discuss it, at least in a footnote.
- Line 265: The authors write “we cannot test how well RealCause fits the interventional distributions P (Y | do(T =t), w)”. If we wanted to realistically evaluate causal estimators, shouldn’t we have benchmarks that accurately capture the interventional distributions?
- Line 283: I would like the authors to clarify why they focus only on a single hyper-parameter.


**Clarity:**

Some points where the writing could have been better:
- Line 96: The authors write “This is probably the most common method for evaluating causal estimators. However, it has several disadvantages”. Some citations would have been useful here.
- Line 122: The authors refer to RCTs. I assume that they mean “randomized control trials”, however the abbreviation is not defined previously.
- Figure 1: This figure could have 4 sub-captions instead of 2. Also, is KDE really useful here? It shows values of Y being lower than zero. I think the histogram would have been enough.
- Line 208: As I mentioned previously, a discussion about the datasets (what are the applications, what are the available features, what is the treatment/effect) would have been useful before discussing the realism of the proposed benchmarking process.
- Line 261: PEHE is mentioned here for the first time but, if I am correct, it is not defined anywhere.

**Correctness:**

I did not get a completely clear understanding of the paper so it’s hard for me to evaluate its correctness.

**Documentation:**

The authors present sufficient detail about their implementation in the main body of the paper and in the appendix. Their source code is publicly available on GitHub and they provide documentation with sufficient information on how to use the code in the repository.

**Ethics:**

Does not apply.

**Relation To Prior Work:**

See “Weaknesses”.

**Summary And Contributions:**

The authors start by providing a brief overview of the basics in causal inference. Then, they discuss the main approaches for the evaluation of causal estimators appearing in prior work and they present the details of their proposed methodology based on fitting generative models using real data. In the experimental part, they first evaluate how well their models fit a set of datasets performing statistical tests. Next, using these generative models, they evaluate the performance of certain causal estimators.

---

> ### Author Response · Authors · 2021-07-11
> **Response**
>
> Because we want these generated datasets to be realistic, it is very important that we test that. As you point out, we spend a lot of space on this in Section 5 (not 6) to show that our models fit the real data well. Imagine that all the other generative models that people fit to datasets in causal inference (most of them are much less flexible) are in a pool. Our tests in Section 5, would rule out almost all of those as realistic. We’ll do our best to make it more clear that the main point of Section 5 is to test how realistic these models are.
>
> “Is just fitting the datasets good enough to claim that this generative process provides a realistic causal model?” -
> As you allude to, the answer is “no”. Please see our “Limitations” paragraph for more on this. The main extra step we took on top of just testing the observational distributions is testing the how well these generative models match the true causal effects of popular simulated synthetic datasets (see Table 3).
>
> “If we wanted to realistically evaluate causal estimators, shouldn’t we have benchmarks that accurately capture the interventional distributions?” -
> Yes, but this is impossible to do on real data due to the fundamental problem of causal inference (we don’t observe the counterfactuals). The best we can do is test how well we fit the interventional distributions on synthetic datasets (see Table 3) and how well we fit the observed distributions on real datasets (see Section 5).
>
> “It is straightforward to extend RealCause to these more complex settings” -
> For example, in the unobserved confounders setting, you just don’t expose some confounders to the user. Or if you wanted some covariates to not be confounders, we’d still give them to the user, but we wouldn’t give them as inputs to our generative models that generate T and Y. Many other extensions like this are possible; these are just two.

---

### Decision · Program_Chairs · 2021-07-26

**Decision:**

Reject

**Comment:**

Reviewers agree that this dataset is addressing an important problem and is timely. Particularly, using realistic dataset to generate causal benchmark and validating it was well-appreciated. However, reviewers agree that the paper would make a bigger impact if it is revised to be more clear and focused. For instance, a reviewer requests for providing better motivations for using tunable unknowns, causal effect strength and heterogeneity. Also, please consider removing the less developed/unused-in-analysis part (e.g., tunable knobs) to appendix to improve the focus of the paper. We recommend authors to incorporate many aspects pointed out by reviewers and resubmit to the 2nd round.